# Primordial neon and the deep mantle origin of kimberlites

Andrea Giuliani [1,2,8] ✉, Mark D. Kurz[3], Peter H. Barry [3], Joshua M. Curtice[3], Finlay M. Stuart[4], Senan Oesch[1], Quentin Charbonnier[1], Bradley J. Peters [1], Janne M. Koornneef [5], Kristoffer Szilas[6] & D. Graham Pearson [7]

The genesis of kimberlites is unclear despite the economic and scientific interest surrounding these diamond-bearing magmas. One critical question is whether they tap ancient, deep mantle domains or the shallow convecting mantle with partial melting triggered by plumes or plate tectonics. To address this question, we report the He-Ne-Ar isotopic compositions of magmatic fluids trapped in olivine from kimberlites worldwide. The kimberlites which have been least affected by addition of deeply subducted or metasomatic components have Ne isotopes less nucleogenic than the upper mantle, hence requiring a deep-mantle origin. This is corroborated by previous evidence of small negative W isotope anomalies and kimberlite location along age-progressive hot-spot tracks. The lack of strong primordial He isotope signatures indicates overprinting by lithospheric and crustal components, which suggests that Ne isotopes are more robust tracers of deep-mantle contributions in intraplate continental magmas. The most geochemically depleted kimberlites may preserve deep remnants of early-Earth heterogeneities.

Noble gases (He, Ne, Ar, Kr, Xe) are robust tracers of the origin of mantle-derived magmas and their study has played a pivotal role in understanding Earth accretion and evolution[1–3]. Intra-plate magmas originating from the lower mantle (e.g., many ocean island basalts or OIBs, and continental flood basalts) show elevated primordial to radiogenic/nucleogenic isotope ratios of He and Ne (i.e., high $^3He/^4He$ and low $^{21}Ne/^{22}Ne$) compared to upper mantle magmas (mid-ocean ridge basalts or MORBs). These observations imply an origin in the relatively undegassed deep mantle, which is enriched in primordial volatiles[1–5]. Variations in $^{129}Xe$ normalised to a non-radiogenic Xe isotope (e.g., $^{130}Xe$), where $^{129}Xe$ is produced by radioactive decay of $^{129}I$ within ~100 Myr of Earth formation, further point to separation between these lower and upper mantle sources early in Earth's history[3]. The exact mechanism leading to the preservation of primordial noble gas signatures in the deep Earth remains contentious. Proposals range

from storage in the core[6–8], dense Fe-rich piles derived from magma ocean crystallisation[9] or simply less processed mantle material[10], perhaps combined with migration and storage of primordial noble gases into geochemically depleted mantle rocks[11]. Contributions from geochemically depleted and deeply subducted material to the sources of deep-mantle plumes is supported by combination of noble gas isotopes with other geochemical tracers, such as the radiogenic isotopes of lithophile elements (Sr, Nd, Hf, Pb) in several OIBs[7,12–14].

The lack of convective homogenisation of primordial noble gases within Earth's mantle makes He and Ne isotopes powerful tracers of the origin of magmas in the deep mantle[1–3]. For example, He isotope ratios substantially higher than the MORB value ($^3He/^4He = 8 \pm 1 \times$ Ra where Ra indicates the atmospheric ratio of $1.384 \times 10^{-6}$[1,15]) support a deep mantle origin for continental magmas associated with large igneous provinces (LIPs)[4,5,11]. Neon isotopes in LIP magmas, although examined

[1]Department of Earth and Planetary Sciences, ETH Zurich, Zurich, Switzerland. [2]Earth and Planets Laboratory, Carnegie Institution of Science, Washington, DC, USA. [3]Woods Hole Oceanographic Institution, Woods Hole, MA, USA. [4]Scottish Universities Environmental Research Centre, East Kilbride, UK. [5]Vrije Universiteit Amsterdam, HV Amsterdam, The Netherlands. [6]Department of Geosciences and Natural Resource Management, University of Copenhagen, Copenhagen, Denmark. [7]Department of Earth and Atmospheric Sciences, University of Alberta, Edmonton, AB, Canada. [8]Present address: Carnegie Institution of Science, Washington, DC, USA. ✉e-mail: agiuliani@carnegiescience.edu

less frequently because their measurement is analytically more challenging due to low Ne concentrations, are less nucleogenic than MORBs. These results are consistent with a deep mantle contribution in the genesis of OIBs[5,16,17]. In some cases, deep-mantle Ne isotopic signatures do not coincide with high $^3He/^4He$ reflecting mixing with deeply subducted material[18], upper convective mantle[19,20] or sub-continental lithospheric mantle[21,22].

Helium and Ne isotopes are ideal candidates to investigate the potential occurrence of deep and ancient material in the sources of magmas, including kimberlites, where such contributions are contentious (see below). Kimberlites are carbonate-rich magmas, which are considered to tap similar mantle sources as those of OIBs, based on overlapping Sr-Nd-Hf isotope compositions[23,24]. Simple petrological considerations, including the instability of carbonates in the deep lithosphere combined with the low degrees of melting (<1%) to generate the observed elevated concentration of mantle-incompatible trace elements, firmly rule out an origin of these melts in the lithospheric mantle[25]. Being emplaced on the stable nuclei of continents (i.e. cratons) since at least 2 Ga[26,27], kimberlites provide a complementary and temporally more extensive perspective on the origin of compositional variations in the convecting mantle than OIBs.

Geographic correspondence between the peripheral zones of large low-shearwave velocity provinces (LLSVPs) above the core-mantle boundary and the reconstructed location of the majority of Phanerozoic kimberlites at the time of eruption has previously been used as evidence to suggest a deep mantle origin for kimberlites[28,29]. In this context, the LLSVP margins represent the loci of mantle plume generation including the sources of several OIBs and LIPs[30]. This view is supported by the spatio-temporal association of some kimberlites with LIPs[31]. Additional models of kimberlite generation entail partial melting of upper convecting mantle sources in response to extensional tectonics[32] or small-scale convection[33]. An upper mantle origin might be supported by temporal correspondence between kimberlite formation and episodes of supercontinent reorganisation[27,32], hence underscoring that no single trigger of kimberlite magmatism exists[25].

The geochemical record of kimberlites similarly includes inconclusive evidence supporting a deep and ancient, rather than shallow mantle origin of these magmas. Long-lived radiogenic isotope systems ($^{87}Rb/^{87}Sr$, $^{147}Sm/^{143}Nd$, $^{176}Lu/^{176}Hf$) indicate that the majority of kimberlites tap a long-lived (>2 Ga) common mantle component that is equivalent to the PREMA (PREvalent MAntle) component documented in OIBs[14] and is potentially, but not necessarily stored in the deep mantle[23,34,35]. The short-lived $^{182}Hf$-$^{182}W$ system seems to support a deep and/or ancient origin of kimberlites based on very small negative anomalies documented in some[36], but not all kimberlites[37]. However, similar anomalies have so far not been detected in the short-lived $^{146}Sm/^{142}Nd$ system[36,38]. Previous analyses of He and Ne isotopes provide inconsistent and therefore inconclusive results with respect to the origin of kimberlites. While non-nucleogenic, plume-like Ne isotope compositions combined with radiogenic He isotopes (<8 Ra) were reported for olivines from the Udachnaya-East kimberlite in Siberia[22], very high $^3He/^4He$ (up to 27 Ra) and MORB-like He isotope values were documented in olivine from two kimberlite provinces in western and southern Greenland (Sarfartoq and Pyramidefjeld, respectively)[39]. In contrast, olivine grains from the Holocene Igwisi Hills kimberlite in Tanzania have relatively low $^3He/^4He$ (-5 Ra)[40].

The goal of this contribution is to provide additional constraints on the genesis of kimberlites by exploring the isotope systematics of noble gases (He, Ne, Ar) in kimberlites worldwide. We present He, Ne and Ar isotope compositions of fluids trapped in 35 olivine separates from 18 kimberlites around the globe with ages ranging from Neoproterozoic (~600 Ma) to Cenozoic (14 kyr), including kimberlites from Udachnaya-East and Sarfartoq (Supplementary Data 1). To understand the origin of olivine-hosted fluids and identify potential contributions from crustal components, noble gas data are supplemented by fluid inclusion petrography and determinations of trace element concentrations and Sr-Nd isotope ratios in the residues of olivines that were crushed in vacuo for noble-gas isotope analysis (see 'Methods'). These analyses are combined with trace element and Sr-Nd-Hf isotope determinations of bulk kimberlite samples from which the olivines were extracted. This approach constrains the roles of source variations, including contributions from ancient lower-mantle and deeply subducted components, interaction with metasomatised lithospheric mantle and crustal contamination in the genesis of kimberlites.

## Results
### He-Ne-Ar isotopes of olivine in kimberlites
The criteria for sample selection includes wide geographic (9 cratons in 4 continents) and temporal coverage (between 582 Ma—Finland Pipe 1—and 14 kyr—Igwisi Hills), limited olivine alteration, and existing noble-gas data (see Supplementary Data 1 for details). The Victor kimberlite (Canada) was included because it is located along the Great Meteor hotspot track[41] while Letseng (Lesotho), Karowe (Botswana) and Monastery (South Africa) are known to contain sub-lithospheric diamonds[42], potentially suggesting very deep origins[43]. The kimberlites were also selected to reflect geochemical variations that suggest input from various components including metasomatised lithospheric-mantle wall rocks in micaceous kimberlites from Sierra Leone (Koidu and Tonguma[44]) or subducted material in those from Lac de Gras (Boa and Leslie[45]) in Canada.

All the kimberlites generally contain abundant macrocrysts (>1 mm) and microcrysts of olivine (Supplementary Fig. S1) and other less abundant phases (e.g., phlogopite, garnet, clinopyroxene, ilmenite) in a fine-grained groundmass dominated by carbonates and/or serpentine with variable spinel, apatite and perovskite. Regardless of size, olivine is typically zoned between mantle-derived xenocrystic cores and magmatic rims (see details in ref.[46]). Fluid inclusions are small (generally <10 μm in size) but abundant and arranged in trails and elongated swarms that cut across olivine zonation (Fig. 1). These inclusions typically host dominant Ca-Mg and alkaline carbonates[47,48]. They are clearly secondary in origin, in that they formed via fluid infiltration after olivine crystallisation at sufficiently high temperatures (e.g., ≥400–500 °C) to allow olivine annealing and entrapment of residual fluids.

To examine the composition of trapped noble gases, olivine grains were separated from crushed kimberlite samples (Supplementary Fig. S2). Noble gas isotope analysis was performed by crushing in vacuum at Woods Hole Oceanographic Institute (WHOI) and Scottish Universities Environmental Research Centre (SUERC) ('Methods'). Helium isotopes are systematically more radiogenic than the MORB composition with $^3He/^4He$ values below 6 Ra (Fig. 2; Supplementary Data 2-3). Analyses undertaken at WHOI and SUERC yield consistent results (Supplementary Fig. S3). $^3He/^4He$ values show a broad inverse relationship with $^4He$ contents (Fig. 2a) indicating that the relative abundance of $^4He*$ (i.e. $^4He$ in-grown or implanted after kimberlite emplacement) profoundly impacts $^3He/^4He$ (see discussion below) and is dominant in the oldest kimberlites. Helium isotopes indeed show a broad relationship with age, where Neoproterozoic and Cambrian kimberlites exhibit the lowest $^3He/^4He$ (≤2.3 Ra; Fig. 2b), including all samples from western Greenland. These results are in stark contrast with those of Tachibana et al.[39] who detected very high $^3He/^4He$ (up to 27 Ra) in kimberlites from Sarfartoq, western Greenland. Conversely, our analyses of Udachnaya-East and Igwisi Hills olivine are entirely consistent with those of Sumino et al.[22] and Brown et al.[40], respectively. The He isotope compositions of Mesozoic and Cenozoic kimberlites are broadly similar to those measured in Cenozoic alkaline mafic lavas from continental intraplate settings (5.9 ± 1.2 Ra[49]; Fig. 2a) with the exception of highly to moderately micaceous kimberlites (Koidu-Tonguma and Kimberley, respectively) which exhibit lower $^3He/^4He$ (<4 Ra). It is noteworthy that mantle xenoliths entrained by

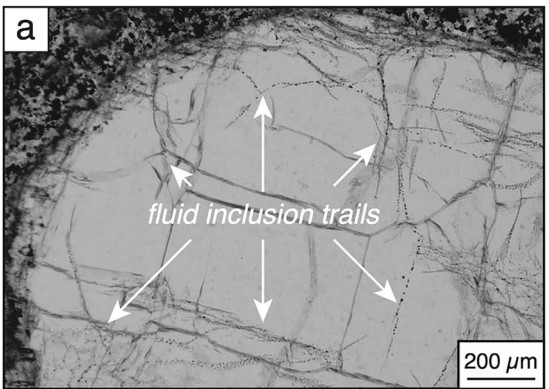
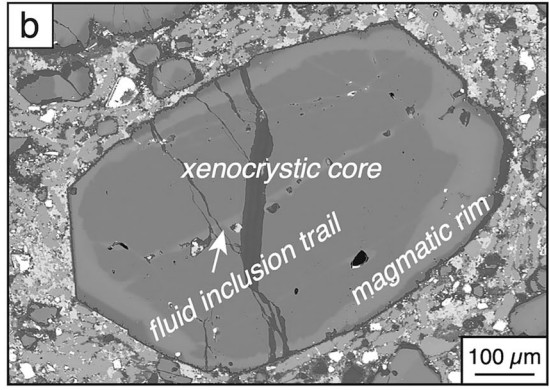

**Fig. 1 | Representative images of trails of secondary fluid inclusions in olivine.** **a** Optical image of fluid inclusions in a olivine macrocrysts from Kimberley (Kimberley, South Africa). **b** Back-scattered electron SEM image of zoned olivine phenocryst from Internationalnaya (Siberia, Russia). Note that the fluid inclusion trails are associated with linear domains of recrystallised olivine that cut across the xenocrystic olivine core.

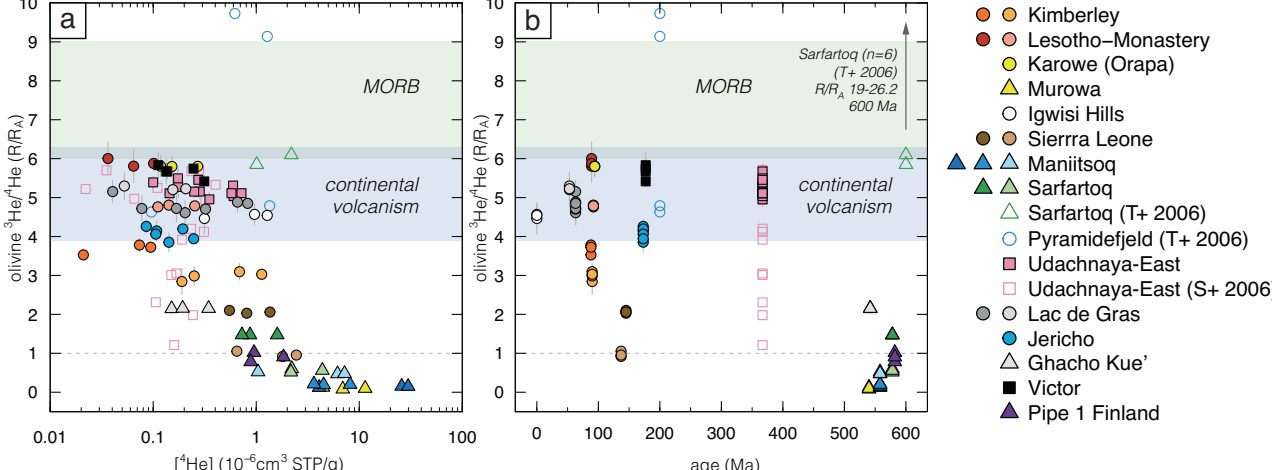

**Fig. 2 | He isotope compositions of olivine in kimberlites.** $^3$He/$^4$He (expressed as ratio R normalised to the atmospheric ratio Ra) *versus* (**a**) $^4$He concentrations in olivine and (**b**) age of kimberlite emplacement for the samples from this study (only data acquired in Woods Hole), Sumino et al.[22] (S + 2006) and Tachibana et al.[39] (T + 2006). Each symbol represents analyses of the same sample. Error bars represent 1σ uncertainties. The $^3$He/$^4$He ranges of MORBs unaffected by plume contributions[1,15] and intracontinental alkaline mafic magmas[49] are shown for comparison. The He isotope data of Igwisi Hills olivine from Brown et al.[40] completely overlap with our results.

kimberlites worldwide (southern Africa, Siberia, Canada[50,51]) show the same range in He isotopes (0.05 to 6 Ra) as kimberlites in this study, except for a less radiogenic dunite xenolith from Udachnaya-East with $^3$He/$^4$He (9.8 Ra) above the MORB range.

In contrast to He, the isotopes of Ne vary from less to more nucleogenic than the reference MORB value[52,53] (Fig. 3). All Ne isotope ratios except for two measurements are resolvable (>2 × standard deviation) from the air composition, clearly indicating that mantle Ne is present in the majority of olivines. Victor and Udachnaya-East exhibit Ne isotope compositions that are less nucleogenic than MORBs with $^{21}$Ne/$^{22}$Ne$_S$ (extrapolated to $^{20}$Ne/$^{22}$Ne = 12.5) of 0.046−0.051 (n = 3) and 0.051−0.055 (n = 8), respectively (where such extrapolation is justified by the ubiquitous occurrence of air-related Ne in olivine). Our Udachnaya-East data are similar, although marginally more nucleogenic than those reported by Sumino et al.[22] for olivine separates from the same locality, resulting in a lower slope for the linear regression through our data in a $^{20}$Ne/$^{22}$Ne-$^{21}$Ne/$^{22}$Ne plot (Fig. 3). These data point to the occurrence of lower-mantle Ne in these kimberlites, as discussed below. $^{40}$Ar/$^{36}$Ar are also typically resolvable from the air composition (298.6) and range up to ~11,000 in micaceous kimberlites from Koidu with high values recorded in other kimberlites (e.g., up 8000 in Udachnaya-East; Supplementary Data 2). Linear correlations between

$^{40}$Ar/$^{36}$Ar and ratios of primordial isotopes (i.e. not affected by radiogenic/nucleogenic processes: $^{20}$Ne/$^{22}$Ne and $^3$He/$^{36}$Ar; Supplementary Fig. S4) in multiple analyses of the same sample indicate a predominant mantle origin for Ar including variable contribution from radiogenic $^{40}$Ar.

**Trace element concentrations and Sr-Nd isotopes in olivine**

Noble gas data provide robust evidence for a mantle origin of the fluids trapped in olivine. However, radiogenic helium is clearly present, and the noble gases do not rule out crustal contributions, which is possible considering the secondary origin of fluid inclusions in kimberlitic olivine. To address this issue, the trace element and Sr-Nd isotope composition of crushed olivine were measured after sample dissolution ('Methods'). The rationale behind this approach is that the concentrations of elements strongly incompatible in olivine, including Rb, Sr, Sm, Nd, Th and U, are completely controlled by the composition and relative abundance of fluid inclusions – solid inclusions in olivine being scarce and dominated by spinel, which is depleted in these elements. This inference is confirmed by the comparison of laser ablation and solution-mode ICP-MS analyses of mantle olivine containing secondary fluid inclusions, which show eight and seven times higher concentrations in solution-mode analyses for Sr and U, respectively[54].

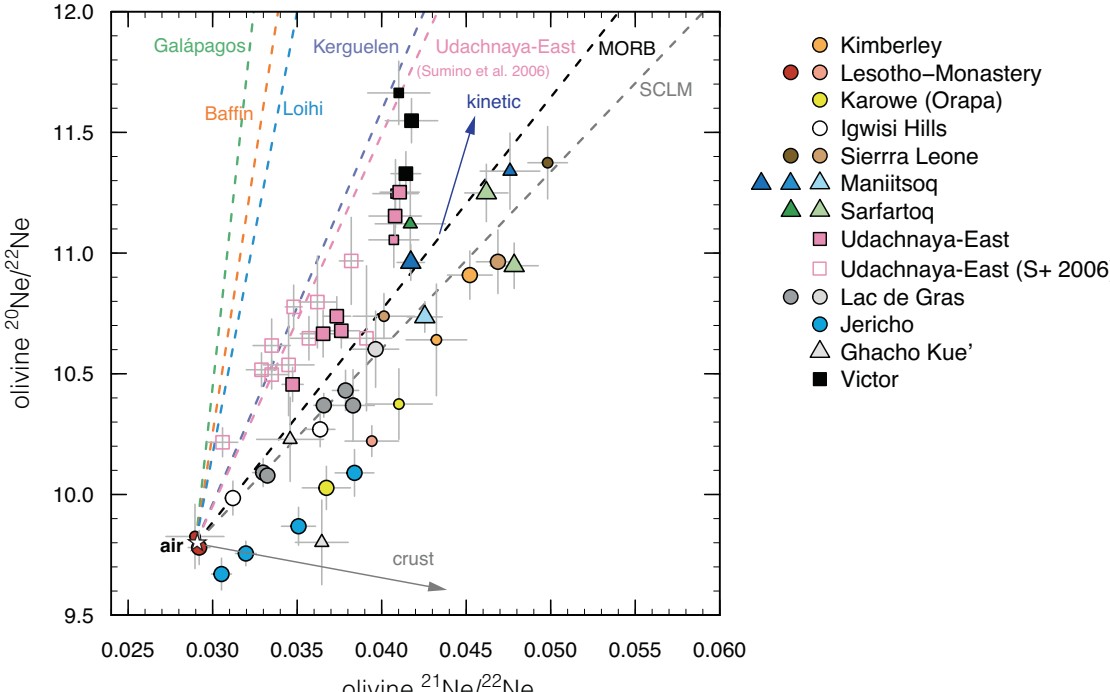

**Fig. 3 | Ne isotope compositions of olivine in kimberlites.** $^{20}$Ne/$^{22}$Ne versus $^{21}$Ne/$^{22}$Ne in olivine from this study and Sumino et al. [22] (S + 2006). Each symbol represents analyses of the same sample, with smaller symbols indicating $^{20}$Ne counts <1 × 10$^{-11}$. Error bars represent 1σ uncertainties. The dotted curves show regressions through representative deep-mantle plumes [Galapagos[73], Baffin[16], Lohi[75,98], Kerguelen[1]], previous data for the Udachnaya-East kimberlite[22], mid-ocean ridge basalts [MORB[52,53,63]] and sub-continental lithospheric mantle xenoliths [SCLM[94]]. The grey arrow labelled 'crust' indicates the approximate trajectory of crustal Ne[60], while the blue arrow labelled 'kinetic' represents the direction of kinetic fractionation from typical MORB compositions[53]. Note that regressions through the Jericho and Karowe data points do not intersect air, which probably indicates trapping of a "fractionated air" component consisting of a mixture of air-Ne and crustal Ne (see 'Methods').

Crushed olivines show high concentrations of compatible elements (e.g., Ni = 1950–2970 ppm; *n* = 21/22, i.e. one analysis with lower Ni) and, as expected, low concentrations of incompatible trace elements (e.g., Sr = 1.3–6.1 ppm; Ce = 0.12–0.98 ppm; U = 2–21 ppb; *n* = 15/22; Supplementary Data 3). Exceptions include olivine separates from Boa (Sr = 75–78 ppm; U = 171-224 ppb; *n* = 2), Igwisi Hills (6.8 ppm; 41 ppb), Ghacho Kué (18 ppm; 27 ppb) and Maniitsoq (≤33 ppm; 35–37 ppb; *n* = 3) where higher contents of Sr, U and other incompatible trace elements are consistent with optical observations of contamination by groundmass material attached to the olivine grains (Supplementary Fig. S1). Compared to bulk rock analyses of the same samples, the olivine separates show broadly similar primitive-mantle normalised patterns of incompatible trace elements although at much lower absolute values (Fig. 4). These characteristics argue for fluid inclusions exerting a dominant control on the trace element budgets of the majority of incompatible elements measured in olivine residues. Positive anomalies of Pb and Sr and flat HREE profiles in some olivine separates compared to negative Pb and Sr anomalies and negatively sloping REE in bulk kimberlites probably stem from localised crustal contribution to the fluids trapped in olivine[55]. Age-corrected Sr and Nd isotopes of the olivine separates are generally indistinguishable from those of bulk kimberlites (Fig. 5 and Supplementary Fig. S5). Exceptions include Victor and, to a lesser extent, Udachnaya-East, where higher $^{87}$Sr/$^{86}$Sr in olivine point towards some crustal contamination of the trapped fluids; and Jericho, where the lower $^{87}$Sr/$^{86}$Sr of olivine rather suggest some crustal contribution to the bulk sample. These data indicate that olivine trapped mostly pristine magmatic fluids with crustal contamination evident in only some cases.

## Discussion

The Udachnaya-East and Victor kimberlites have less nucleogenic Ne isotope compositions compared to the upper mantle source of MORBs (Fig. 3). These data point to lower-mantle contributions for these kimberlites potentially related to plumes from the core-mantle boundary. However, the lack of high $^3$He/$^4$He in Udachnaya-East and Victor (Fig. 2), despite their plume-like Ne isotopic compositions, is intriguing. The absence of plume-like $^3$He/$^4$He in West Greenland is also an interesting finding, even though five kimberlites from two adjacent fields were examined, including Sarfartoq where high $^3$He/$^4$He values had been previously reported[39]. The observed relationships between $^3$He/$^4$He and both $^4$He and age (Fig. 2) suggest a likely addition of $^4$He* either in-grown or implanted into the olivine inclusions, at least for the Cambrian and Neoproterozoic samples. Calculations of $^4$He* ingrowth and implantation are highly model-dependent and hence inconclusive (see 'Methods' for details of calculations and related discussion). These data, however, do demonstrate the challenges of interpreting He measurements in ancient Th-U-rich rocks. The measured $^3$He/$^4$He therefore represent minimum values for the kimberlite mantle sources in the Cambrian and Neoproterozoic samples, which might have been further lowered by crustal contamination and interaction with the lithospheric mantle as discussed below. The role of $^4$He* ingrowth and implantation in the younger samples is probably more limited, and effectively negligible in the Holocene kimberlites from Igwisi Hills. Below, the full array of data acquired in this study is employed to assess processes which might have affected noble gases in kimberlites and their olivine during magma ascent and emplacement, followed by potential variations in kimberlite source compositions.

### Noble gas modification during kimberlite ascent and emplacement

During ascent to the surface, kimberlites exsolve abundant volatiles, a process which probably governs their very fast ascent and promotes their ability to transport xenoliths up to several decimetres in size. The

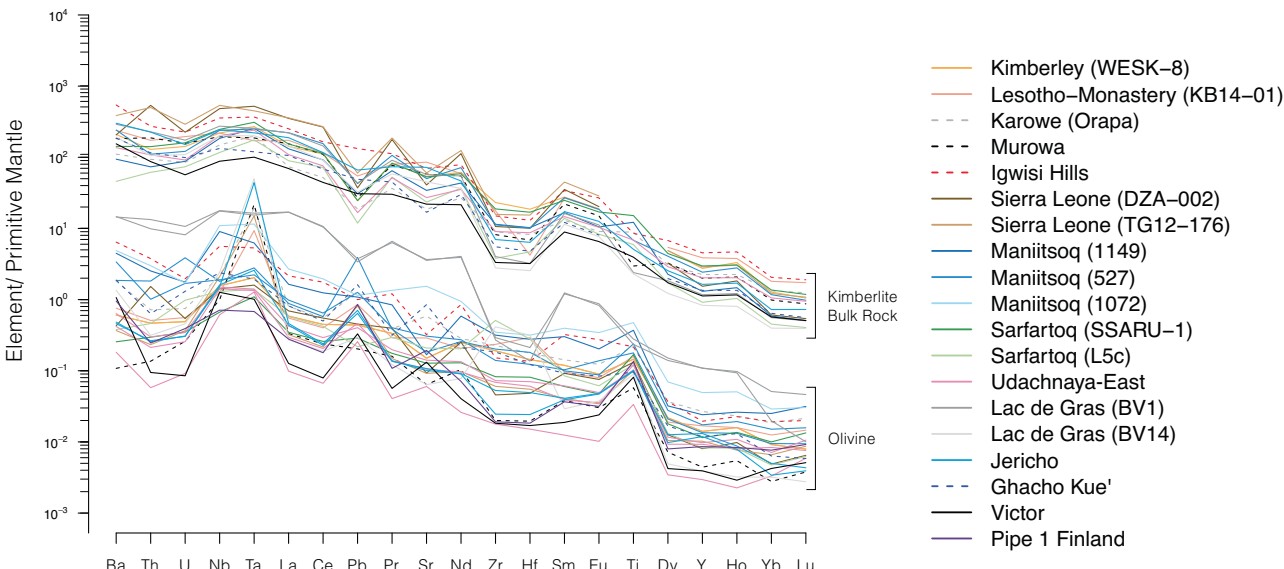

**Fig. 4 | Primitive-mantle normalised trace element diagrams of olivine separates and bulk kimberlite rocks.** Positive anomalies for Ta (±Nb) and Ti in olivine probably indicate inclusions of ilmenite (±spinel) in some olivine separates (see Supplementary Fig. S1). Positive anomalies of Pb and Sr coupled with flat HREE patterns are consistent with variable crustal contamination in some olivine separates. Beyond these differences, the patterns of olivine separates show similar shapes to those of their bulk kimberlite samples, which underpins a magmatic origin for the fluids trapped in olivine.

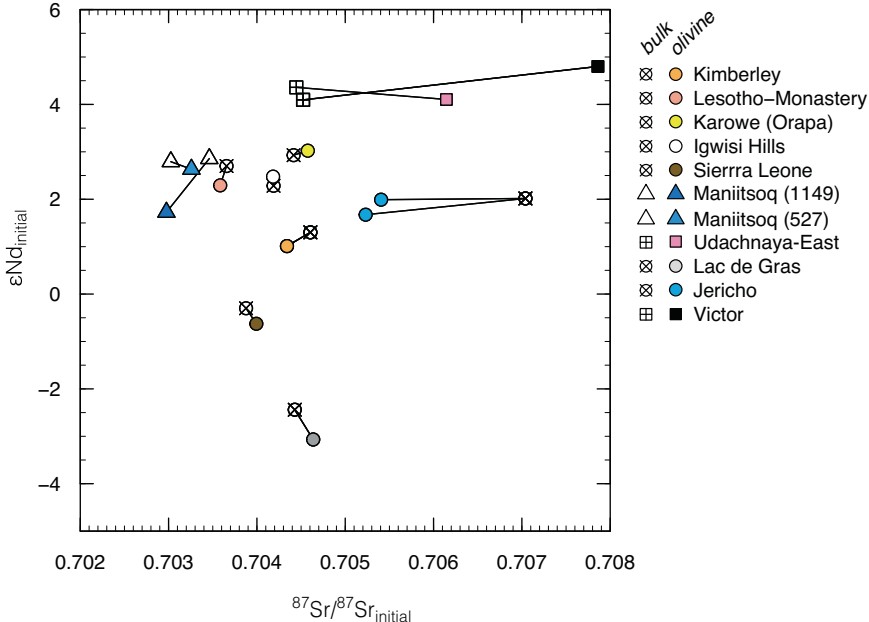

**Fig. 5 | Comparison of Sr and Nd isotopes in olivines and kimberlites.** Sr and Nd isotopes are corrected for radiogenic ingrowth using kimberlite emplacement ages (Supplementary Data 1), and measured Rb/Sr and Sm/Nd ratios (Supplementary Data 3). Straight connectors link olivine compositions to those of bulk kimberlite samples ('bulk') from which the olivine grains were separated. The isotopic compositions of olivine are assumed to reflect those of trapped fluid inclusions.

contrasting solubilities of noble gases in melts and volatile-rich fluids affect their relative abundances in the fluid inclusions trapped by olivine as suggested by the direct correlations observed between log($^4$He/$^{40}$Ar*) and log($^4$He/$^{21}$Ne*) (Fig. 6). $^4$He/$^{21}$Ne* values exceeding the mantle production ratio (2.2 × 10$^7$ [56]) further point to substantial contribution of post-crystallisation $^4$He* in olivine from Neoproterozoic kimberlites. The low He/Ne and He/Ar of olivine suggest variable He loss after kimberlite emplacement. Alternatively, the trapped fluids could represent an exsolved fluid phase, rather than residual melts[47,48], due to lower solubility of He compared to Ne and Ar in hydrous fluids relative to silicate and carbonate melts[20,57,58].

Beyond exsolving fluids, kimberlites variably interact with crustal rocks during emplacement[59]. Crustal contamination, perhaps via mixing with crustal fluids, is evident in the higher Sr isotope compositions of olivine versus bulk rocks from Victor and Udachnaya-East (Fig. 5). It is also supported by positive Pb and Sr anomalies in the trace element patterns of these olivines (Fig. 4). Crustal contamination introduces strongly radiogenic He and nucleogenic Ne in mantle-derived magmas due to the scarcity of $^3$He and $^{22}$Ne in the crust. In addition, the rate of $^4$He* production in contaminated magmas is high due to high abundances of U and Th in continental crust[60]. The isotopes of He are therefore more substantially affected by crustal

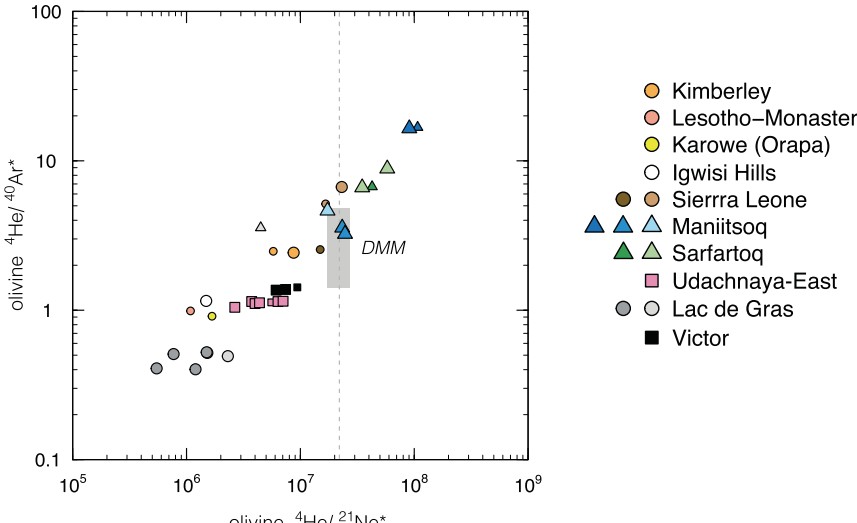

**Fig. 6 | Extent of He loss in fluids trapped by olivine.** $^4He/^{21}Ne^*$ versus $^4He/^{40}Ar^*$ in olivine where $^{21}Ne^*$ and $^{40}Ar^*$ represent nucleogenic Ne and radiogenic Ar, respectively, that is corrected for air contribution and assuming $^{20}Ne/^{22}Ne = 12.5$ for $^{21}Ne^*$. Each symbol represents analyses of the same sample, with smaller symbols indicating $^{20}Ne$ counts $<1 \times 10^{-11}$. The vertical dotted lines represent the mantle (and crustal) production rate of $2.2 \times 10^{7}$[56]. DMM represent the composition of undegassed MORBs[1].

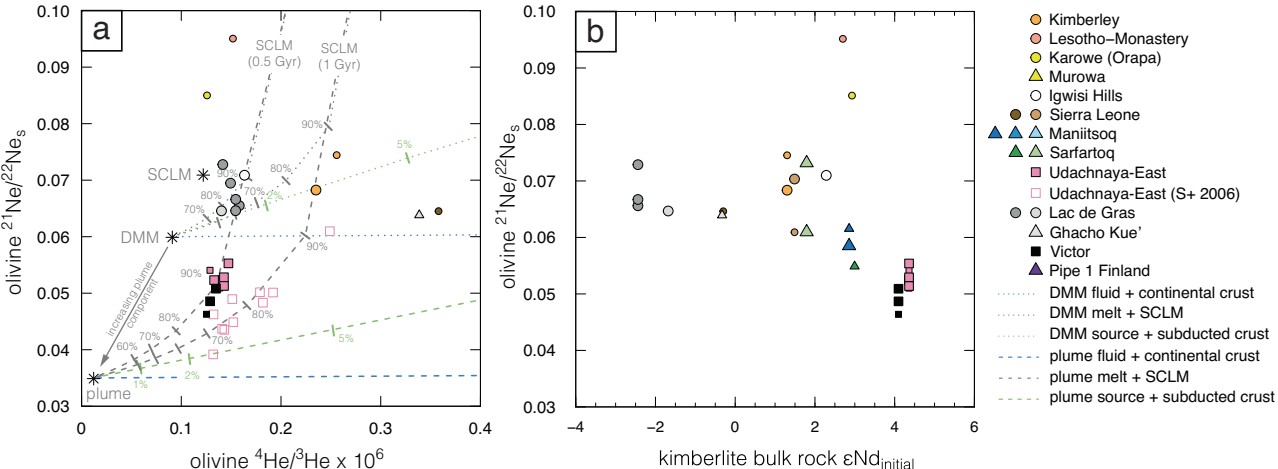

**Fig. 7 | Comparison of Ne, He and Nd isotopes in kimberlites.** Olivine $^{21}Ne/^{22}Ne_S$ versus (**a**) $^4He/^3He$; and (**b**) bulk-rock Nd isotopes in the samples from this study and Sumino et al. [22] (S + 2006). Each symbol represents analyses of the same sample, with smaller symbols indicating $^{20}Ne$ counts $<1 \times 10^{-11}$. $^{21}Ne/^{22}Ne_S$ is the extrapolation of $^{21}Ne/^{22}Ne$ to $^{20}Ne/^{22}Ne = 12.5$. Panel **a** shows mixing trajectories between a plume component from the lower mantle (either 'source', 'melt' or a 'fluid' exsolved therefrom) with the isotopic composition of the Baffin picrites ('plume'[6,16]); or an upper mantle component with the isotopic composition of MORBs ('DMM'[1]); and subducted oceanic crust ('subducted crust' mixed in the source), metasomatised sub-continental lithospheric wall rocks ('SCLM' assimilated during ascent) and continental crust contaminant ('crust' contributing to the exsolved fluid). The % values next to the mixing curves indicate the amount of crustal or lithospheric mantle component added. Details of the endmember compositions employed for mixing are in 'Methods'. Note that while mixing models involving SCLM wall rocks intersect the compositions of olivine in kimberlites, these models require unrealistically large contributions (generally >70%) from the lithospheric mantle. The high $^{21}Ne/^{22}Ne_S$ calculated for Karowe and Letseng (Lesotho-Monastery) are probably unrealistic (see 'Methods').

contamination than those of Ne because kimberlite fluids trapped in olivine have low He/Ne compared to the $^4He$-rich continental crust – the latter being approximated by the $^4He/^{21}Ne^*$ crustal production rate[60] (Fig. 6; see also the 'crustal contamination' curve in Fig. 7). The implication is that $^3He/^4He$ in Victor and Udachnaya-East olivine represent minimum values compared to their sources – a hypothesis explored further below.

Whilst crustal contamination is not ubiquitous, a fundamental role of interaction with lithospheric mantle wall rocks is well established for the petrogenesis of kimberlites worldwide[25,46,61]. For example, a combination of mica enrichment, elevated Sr isotopes and chondritic to marginally super-chondritic Nd-Hf isotopes in micaceous kimberlites from Koidu and Tonguma probably indicates an important

contribution from metasomatised, phlogopite-bearing lithologies in the lithospheric mantle that was traversed by kimberlite melts sourced from the convecting mantle[44]. These kimberlites show the lowest $^3He/^4He$ (≤2.3 Ra) among the samples younger than 500 Ma (Fig. 2) with Ne isotopes typical of the SCLM (Fig. 3) and $^4He/^{21}Ne^*$ close to the mantle production rate (Fig. 6). Although it seems possible that $^4He$ is derived in part from implantation from the micaceous groundmass (see 'Methods'), these data suggest that phlogopite-rich lithospheric mantle contributed, at least marginally, to the noble gas and volatile budget of the Koidu and Tonguma kimberlites.

A similar conclusion can be reached for the moderately micaceous kimberlites from Kimberley (Wesselton and Bultfontein) which show broadly similar isotopic compositions to the Koidu and Tonguma

samples for the noble gases and lithophile elements Sr, Nd and Hf (Figs. 2–5; Supplementary Fig. S5) combined with a well-established enrichment in mica of the underlying lithospheric mantle[62]. Mixing models show that the He-Ne isotope composition of the Kimberley kimberlites can be crudely reproduced by combining He and Ne from a melt derived from either an upper mantle source with MORB-like He-Ne isotope compositions[1,63], or a lower-mantle source similar to that of the Baffin picrites[6,11,16], and noble gases from phlogopite-rich lithospheric mantle (Fig. 7; see 'Methods' for details). However, all of these models entail deriving disproportionate amounts of noble gases from the lithospheric mantle (>70–80%), given that kimberlite melts are likely to have at least one or two orders of magnitude higher concentrations of noble gases than metasomatised lithospheric mantle lithologies. Therefore, unless either the lithospheric mantle hosts substantially larger amounts of He and Ne than those measured in mantle xenoliths or we have grossly overestimated the concentrations of noble gases in kimberlites, interaction with lithospheric mantle rocks can only have a limited effect on the noble gas isotope composition of kimberlites.

## Noble gas constraints on the kimberlite source

The alternative explanation for the He-Ne isotope signature of the Kimberley kimberlites is a contribution from subducted crustal material, a model previously invoked to explain the Sr-Nd-Hf and S isotope systematics of these kimberlites[23,64]. The oceanic crust loses all of its He and most of its Ne budget during subduction[65,66] but experiences substantial ingrowth of radiogenic He and nucleogenic Ne during mantle residence (see 'Methods'). Mixing models show that addition of subducted material to the convecting mantle can generate He-Ne isotopic signatures that are intermediate between those produced by crustal contamination (substantial decrease of $^3$He/$^4$He at relatively invariant $^{21}$Ne/$^{22}$Ne$_S$ for low degrees of contamination) and interaction with metasomatised lithospheric mantle (moderate decrease in $^3$He/$^4$He with increasing $^{21}$Ne/$^{22}$Ne$_S$; Fig. 7). Mixing trajectories between an upper mantle source (with or without some lower-mantle influence) and subducted oceanic crust (<5%) intersect the composition of the Kimberley kimberlites (Fig. 7). The He-Ne isotopes of the Lac de Gras kimberlites may be similarly explained by adding <2% of subducted crust to an upper mantle source (Fig. 7), which is consistent with the peculiar geochemically-enriched Nd-Hf isotope compositions[45] and the lack of a geodynamic connection with deep-mantle plumes for these kimberlites[25,29]. It is noteworthy that the mixing model requires the subducted crustal material to have higher Ne/He than the convecting mantle source to fit the Kimberley and Lac de Gras data, which is consistent with the more efficient loss of He from subducted materials compared to Ne during subduction via dehydration[66,67] and/or during mantle storage via diffusion.

Olivine separates and bulk-kimberlite samples from Victor and Udachnaya-East exhibit the highest age-corrected Nd isotope compositions (εNd$_i$; Figs. 5 and 7) with values overlapping the evolution curve of the common, moderately geochemically-depleted (PREMA-like) component identified in kimberlites worldwide[35]. Despite some crustal contamination of the trapped fluids based on higher $^{87}$Sr/$^{86}$Sr$_i$ of olivine compared to their bulk rocks (Fig. 5), these kimberlites have experienced minimal contributions from subducted crustal material or interaction with metasomatised lithospheric mantle, which both lower εNd$_i$. We believe these minimal contributions favoured the preservation of plume-like Ne isotopes in these kimberlites (Fig. 7). This is a key finding of this study: i.e. that isotopic signatures of deep and/or ancient mantle are preserved exclusively in the most geochemically-depleted kimberlites and only by isotopic systems least affected by crustal contamination. This conclusion contrasts, at least partly, from observations available for other mantle-derived magmas such as ocean island basalts where Ne-isotope plume signatures are associated with geochemical indicators (e.g., Sr-Pb isotopes) of subducted material in

their sources[18,68,69]. A connection between the Victor kimberlite and a deep-mantle plume is consistent with its location along the Great Meteor hot-spot track[41]. High $^3$He/$^4$He (up to 52 Ra) in sub-lithospheric diamonds from Juina (Brazil) were similarly suggested to be linked to the deep-mantle plume that generated Cretaceous kimberlites in Brazil[70]. Udachnaya-East shows the most negative μ$^{182}$W recorded in kimberlites[36] with olivine in one mantle xenolith and clinopyroxene in three mantle xenoliths from this locality exhibiting $^3$He/$^4$He values marginally above the MORB range[50] and Ne isotopes less nucleogenic than MORBs[71], respectively. These features, as well as unradiogenic He and non-nucleogenic Ne in kimberlite-related fibrous diamonds from the nearby Nyurbinskaya kimberlite[72], point to lower mantle contributions probably related to deep-mantle plumes for the Devonian kimberlites in Siberia, including Udachnaya-East. While we cannot completely rule out kinetic fractionation[53] as an explanation for plume-like Ne isotopes (i.e., vector towards increasing $^{20}$Ne/$^{22}$Ne in Fig. 3), we note that samples with the most geochemically-depleted compositions (i.e. highest Nd isotopes) also appear to have other features supporting a deep and/or ancient mantle contribution, therefore lending support to the plume hypothesis.

Yet, both Victor and Udachnaya-East olivines exhibit moderately radiogenic $^3$He/$^4$He (5.4-5.8 Ra and 5.0-5.7 Ra, respectively) apparently at odds with their low $^{21}$Ne/$^{22}$Ne$_S$. If He and Ne isotopes reflect the secular evolution of a compositionally homogeneous source, production of $^4$He* and $^{21}$Ne* should be tightly linked by radioactive decay of U and Th, as it is commonly the case for young oceanic basalts and their olivines (e.g.[1,52,73]). This is clearly not the case for Victor and Udachnaya-East where 'decoupling' of He and Ne isotopes requires contribution of at least an additional component, perhaps crustal contamination identified in the olivine Sr isotopes (Fig. 5). Modelling shows that this process can lower $^3$He/$^4$He while leaving Ne isotopes largely unaffected (Fig. 7) with Nd isotopes being similarly unmodified from their mantle values due to low solubility of REE in hydrous fluids and moderately low concentrations of LREEs in the continental crust[74]. Crustal contamination is a viable explanation for the relatively low $^3$He/$^4$He only if the $^{21}$Ne/$^{22}$Ne$_S$ of the source was intermediate between the Baffin picrites and MORB values as can be gauged from the mixing trajectories in Fig. 7. Assuming similar $^{22}$Ne (and $^3$He) contents in the sources, higher $^{21}$Ne/$^{22}$Ne$_S$ (and lower $^3$He/$^4$He) of the kimberlite source compared to typical deep-mantle plumes such as Baffin[16], Hawaii[75] or Galapagos[73] is consistent with a more fertile source for kimberlites (or at least more fertile components participating in partial melting) containing higher U and Th concentrations.

## Implications for noble gas signatures in the lithospheric and deep convective mantle

This study shows that kimberlites, which derive from the convective mantle[24,25,27,45], share similar He isotope compositions to the mantle xenoliths they entrain (0.05 to 6 Ra[50,51]). This overlap confirms previous suggestions that noble gases in mantle xenoliths are dominated by input from entraining and/or precursor magmas which infiltrate the lithospheric mantle not long before eruption[51,76]. This view is consistent with the dominant secondary and, therefore, late origin of fluid inclusions in mantle xenoliths as well as abundant evidence of interaction between mantle xenoliths and their transporting media[25,77]. It is further supported by the occurrence of plume-like He-Ne isotope compositions in lithospheric mantle xenoliths transported to the surface by plume-related magmas at Samoa, Hawaii and in south-eastern Australia[78,79]. Although radiogenic and nucleogenic noble gases produced in situ in the lithospheric mantle might contribute to the percolating sub-lithospheric melts[49,51], mass balance calculations presented in this work indicate a limited role for indigenous lithospheric-mantle noble gases in the convecting-mantle melts that traverse the lithospheric mantle.

It is clear that at least some kimberlites, and specifically those least affected by addition of deeply subducted material and/or interaction with metasomatised lithospheric-mantle rocks (e.g., Victor, Udachnaya-East), contain primordial Ne derived from the lower mantle. These results combined with previous noble gas[22], W isotope[36] and Nd-Hf isotope data[35], as well as geodynamic reconstructions[28,29,41], underline a link between some kimberlites and plumes from ancient domains in the lower mantle[43]. It is noteworthy that these kimberlites are not renowned hosts of sublithospheric diamonds, which underscores a potential dichotomy between kimberlite source regions and origin of entrained sublithospheric material, the latter probably sourced from lithologies that underplated continental lithospheric roots[80]. Conversely, it is evident that some kimberlites (e.g., Lac de Gras) are not related to deep-mantle plumes as shown by the noble-gas data presented herein, while the data are inconclusive for other kimberlites (e.g., Kimberley) where subducted crustal material appear to largely influence the isotopes of He and Ne. In these cases, it is currently not possible to unanimously establish whether the sources of these kimberlites are in the upper or lower convecting mantle. Application of noble gas isotope analyses, combined with the analysis of the decay products ($^{129}$Xe, $^{182}$W, $^{142}$Nd) of short-lived isotopes, to kimberlites minimally affected by components derived from subducted material, interaction with metasomatised lithospheric mantle and/or crustal contamination will help elucidate the origin of deep and ancient components in the genesis of kimberlites. This work establishes the examination of geochemically depleted kimberlites as a promising tool to investigate the preservation of ancient mantle heterogeneities in the deep Earth.

A corollary of this study is that Ne isotopes, in samples where mantle contributions can be separated from air contamination, represent more robust tracers of the preservation of early Earth heterogeneities in magmas from the deep mantle compared to He isotopes, especially in continental settings. The likely reason is that plumes from the lower mantle have Ne/He ratios substantially higher than those of upper mantle and crustal contaminants as also noted in some previous studies[19–21,73], combined with the importance of radiogenic helium in the older samples, herein demonstrated using kimberlites. Neon isotope measurements of intraplate continental lavas, for which data are restricted to some LIPs[5,16,17,19,81], therefore represent a new avenue of research to detect potential contributions by deep-mantle plumes or material thereof, especially if combined with petrographic, trace element and Sr-Nd isotope analyses of fluid inclusions in olivine. This approach can be extended to ancient rocks where interpretation of He isotopes is complicated by radiogenic ingrowth and implantation, processes that affect Ne to a much lesser extent.

## Methods

### Noble gas analyses

Offcuts of kimberlite samples were fragmented using either a ring mill or a Selfrag, which employs electrostatic discharges to disaggregate rock samples along grain boundaries. This latter method produces cleaner olivine grains, i.e. with less adhered groundmass. After sieving the crushed material, olivine was separated from the 1–2 mm and 0.7–1.0 mm size fractions using a binocular microscope. Separated olivines were then cleaned using various combinations of diluted nitric acid and acetic acid followed by distilled water and acetone to remove carbonates and other groundmass impurities. This step was followed by further olivine purification by picking out olivine grains with evident contamination by extraneous material. For samples BV-1 (Boa, Lac de Gras) and MPV-04-193 (Ghacho Kué) it was not possible to obtain a clean separately, whereas for samples WESK-8 (Wesselton, Kimberley) and 527 (Maniitsoq, West Greenland) both clean and contaminated separates were obtained and measured for comparison (Supplementary Fig. S3).

The analyses of noble gas (He, Ne and Ar) contents and isotopic compositions were conducted at Woods Hole Oceanographic Institutions (WHOI) with a subset of the same samples, including two contaminated olivine separates, measured at the Scottish Universities Environmental Research Centre (SUERC; He only). In both laboratories, the analyses were conducted exclusively by in vacuo crushing using a magnetically actuated metallic sphere and a hydraulic crusher equipped with a vertical piston, respectively. At WHOI, He, Ne and Ar were measured sequentially on the same sample aliquot with loaded olivine masses ranging between 0.21 g and 1.16 g, but generally >0.6 g. Two crushing steps of 10, 20 or 40 strokes each were generally applied to extract the gases trapped in fluid inclusions. The extracted gases were purified on a fully automated extraction line using two SAES ST707 pellet getters, one held at 300 °C and the other between room temperature and 300 °C. The gases were pre-measured (and split if necessary) using a quadrupole mass spectrometer (QMS) to ensure that gas sample size in the mass spectrometer matched standard size. Following purification and trapping, the noble gases were selectively desorbed from a charcoal trap (helium) and a nude stainless-steel trap (neon and argon). Helium and Ne isotopes were measured using a MAP 215-50 mass spectrometer locally referred to as MS5, and Ar isotopes using a dedicated Hiden QMS. Standards and blanks bracketed the samples as closely as possible. The helium standard tank was produced from MORB glass with a $^{3}$He/$^{4}$He of 8.34 ± 0.05 Ra (calibrated against air using a separate mass spectrometer); error estimates were dominated by reproducibility of standard aliquots, which was typically 1%. Neon and argon were calibrated with air aliquots from a separate tank. Helium blanks were less than $1 \times 10^{-11}$ cc STP $^{4}$He; blanks for $^{20}$Ne and $^{40}$Ar during the course of this study were typically less than $1 \times 10^{-12}$ and $2 \times 10^{-10}$ cc STP, respectively. Blanks are negligible for helium and argon, but neon isotope data are considered to be robust only for $^{20}$Ne $> 1 \times 10^{-11}$ cc STP in the mass spectrometer (approximately 10 times the blank) but are still reported for comparative purposes where $^{20}$Ne is higher than $6 \times 10^{-12}$ cc STP (small symbols in Figs. 3, 6 and 7). The mass spectrometer and extraction line have been described previously[63,73].

For the He analyses at SUERC, approximately 0.30 g of olivine was crushed. The released gas was purified using SAES getters and two liquid-N cooled charcoal traps. Helium concentrations and isotope compositions were measured using a *ThermoFisher* Helix SFT mass spectrometer tuned to the maximum sensitivity. The average blank levels are $2 ± 0.2 \times 10^{8}$ and $5 ± 2 \times 10^{3}$ atoms of $^{4}$He and $^{3}$He, respectively. Sensitivity and mass discrimination were determined by repeated analysis of the HESJ standard revealing a reproducibility of ±0.2%. The crushing procedure releases <0.05% of the total cosmogenic He present in olivine[82] so the analyses presented here are largely unaffected by lattice-hosted He components.

### Olivine and kimberlite trace element and Sr-Nd-Hf isotope analyses

Residues of crushed olivine after the noble gas analyses were employed for trace element and Sr-Nd isotope analyses by solution methods at ETH Zurich. For complete sample characterisation, the trace element and Sr-Nd-Hf isotope compositions of bulk kimberlite samples were also characterised following the same analytical procedure outlined by Fitzpayne et al.[44], which is summarised below. About 200 mg of crushed olivine and 100 mg of powdered fresh chips of kimberlite were digested during 48 h in Teflon bomb at 120 °C using an HF-HNO$_3$ mix (3:1). Residues such as oxides or fluorides were subsequently re-dissolved using *aqua regia* to obtain optically clean solutions. A 10% fraction of each dissolved sample was set aside for trace element analysis and dried down before dilution into 2% HNO$_3$ – 0.005 M HF. A multi-element spike containing Be, In and Bi was added prior to analysis of the solutions, which was undertaken using an Element XR sector field ICP-MS. The isotopes $^{9}$Be, $^{115}$In and $^{209}$Bi were used

as internal standards, and the reference materials BIR-1 (USGS basalt) and BE-N (CRPG basalt) were used for calibration of olivine and bulk kimberlite analyses, respectively. USGS basalt BCR-2 and BHVO-2 were analysed as unknowns for quality control and yielded results consistent with published reference values (Supplementary Data 3).

Strontium, Nd, and Hf were separated from the remaining solutions using ion-exchange column-chromatography methods adapted from Münker et al. [83] and Pin et al. [84]. Procedural blanks for Sr (45 pg), Nd (20 pg), and Hf (30 pg) were all negligible relative to the sample amounts including Sr and Nd in the olivine separates (e.g., Sr loads generally between 5–10 ng). Strontium isotope analyses of olivine and bulk-rock were carried out using a Thermo-Fisher Triton thermal ionisation mass spectrometer (TIMS) and a Neptune MC-ICP-MS, respectively. Instrumental mass bias was corrected by internal normalization to $^{88}Sr/^{86}Sr = 8.37521$ using the exponential law and for the TIMS analyses $^{87}Sr/^{86}Sr$ additionally normalized to a preferred $^{87}Sr/^{86}Sr$ ratio for the NBS SRM987 standard ($^{87}Sr/^{86}Sr = 0.710249$) based on the average measured $^{87}Sr/^{86}Sr$ ratio for SRM987 in the barrel (for TIMS) or session (for MC-ICP-MS) in which the sample was analysed. The USGS basalt BHVO-2, BIR-1 and BCR-2 were analysed as unknowns alongside the olivine samples and returned $^{87}Sr/^{86}Sr$ values within uncertainty of expected values (Supplementary Data 3). Analyses of Nd and Hf isotopes were carried out using a Nu Plasma II multi-collector inductively coupled plasma mass spectrometer (MC-ICP-MS). Instrumental mass bias was corrected by normalization to $^{146}Nd/^{144}Nd = 0.7219$ and $^{179}Hf/^{177}Hf = 0.7325$ using the exponential law. $^{143}Nd/^{144}Nd$ and $^{176}Hf/^{177}Hf$ ratios for unknowns and secondary standards are normalized to La Jolla Nd = 0.511858 and JMC475 = 0.282160, respectively. Analyses of secondary reference materials including the J-Nd solution standard and USGS basalts are consistent with accepted values (Supplementary Data 3). Age corrections were calculated using kimberlite emplacement ages available in the literature (Supplementary Data 1) and $^{87}Rb/^{86}Sr$, $^{147}Sm/^{144}Nd$ and $^{176}Lu/^{177}Hf$ ratios derived from trace element data for the same sample solutions (Supplementary Data 3). $\varepsilon_{Nd}$ and $\varepsilon_{Hf}$ values are calculated relative to the chondritic (CHUR) composition of Bouvier et al. [85]. The Rb, Sm and Lu decay constants are $1.397 \times 10^{-11}$/yr, $6.54 \times 10^{-12}$/yr and $1.865 \times 10^{-11}$/yr, respectively.

### Olivine selection and assessment of sample contamination

Most of the helium data in the literature is derived from measurements of basaltic glasses and crystals in young lava flows; there are few data from older lava flows, and even fewer from kimberlites. Selection of fresh olivine that is not contaminated by serpentine or attached groundmass (Supplementary Fig. S2) appears to have an important control on measured He isotopes. In order to evaluate the influence of groundmass and serpentinization processes, two samples with sufficient material (Wesselton WESK-8; Maniitsoq 527) were divided into relatively "pure" and "contaminated" olivine fractions. These two fractions showed markedly different $^3He/^4He$ between the uncontaminated and contaminated fractions, with the contaminated olivines yielding two to three times lower $^3He/^4He$ ratios (Supplementary Fig. S3 and Supplementary Data 2). It is difficult to definitively attribute this difference to contributions of crustal $^4He$ in hydrothermal fluids that crystallise serpentine in kimberlites[47] or $^4He*$ produced in the U-Th-rich groundmass attached to the contaminated olivine grains. Regardless, the olivine separates used for this study generally contain negligible contamination compared to the contaminated olivines of samples WESK-8 and 527 where olivine was expressly selected to monitor the effect of serpentinisation and contamination by groundmass material.

### Modelling of $^4He*$ ingrowth and implantation

No attempt was made to estimate or correct for the potential effect of cosmogenic $^3He$ since kimberlite emplacement because all the samples come from underground mining activity with the exception of the

West Greenland samples where snow coverage provides screening from cosmic rays for most of the year. Ingrowth and implantation of radiogenic $^4He*$ probably affected measured $^3He/^4He$ and was modelled as follows.

Radiogenic $^4He$ generated in olivine fluid inclusions after olivine formation ($^4He*$) was calculated using the following equation[1]:

$$^4He^* = {}^{238}U \times \left\{ 8 \times \left[ e^{(0.155125 \times t)} - 1 \right] + \frac{7}{137.88} \times \left[ e^{(0.98485 \times t)} - 1 \right] + 6 \times k \times \left[ e^{(0.049475 \times t)} - 1 \right] \right\} \quad (1)$$

Where $^{238}U = U \times 0.9927$, $k = {}^{232}Th/^{238}U$, and U and Th concentrations are from bulk olivine dissolution analyses (Supplementary Data 3); $t$ is the kimberlite emplacement age (Supplementary Data 1); and both $^{238}U$ and $^4He*$ are in mol/g. $^4He*$ produced by the U and Th measured in dissolved olivine exceeds the measured $^4He$ content in 20% of the analyses (n = 5/25; Supplementary Fig. S6). This is probably due to incomplete release of lattice-hosted $^4He$ by crushing[51,86]. In addition, it is likely that $^4He*$ is not fully retained in fluid inclusions due to alpha particle (i.e. $^4He*$) ejection outside fluid inclusions into olivine lattice[87] and loss of $^4He*$ via diffusion and through cracks. Only a fraction of $^4He*$ is retained in the fluid inclusions because alpha particle recoil distance in olivine (15–20 μm) exceeds the typical size (<10 μm) of fluid inclusions (Fig. 1). However, $^4He*$ could have been recoiled from nearby fluid inclusions because fluid inclusions in kimberlitic olivine generally occur in clusters and swarms of tens to hundreds or more inclusions. Hence, for modelling purpose, we arbitrarily assume that 10% of $^4He*$ produced in olivine fluid inclusions (± lattice and material in crack and/or adhering to olivine) was released during crushing, which accounts for alpha particle injection into fluid inclusions from neighbouring fluid inclusions and $^4He*$ diffusion from lattice to fluid inclusions. In this scenario, age-corrected $^3He/^4He_i$ would be similar to measured $^3He/^4He$, except for some olivines showing clear evidence of groundmass contamination (e.g., Boa, Ghacho Kué; Supplementary Fig. S6). Hence, unless a substantially higher amount of $^4He*$ is retained in the fluid inclusions, ingrowth of $^4He*$ does not fully explain the more radiogenic He isotopes of Cambrian and Neoproterozoic kimberlites (Fig. 2).

In alternative, the high U and Th contents of kimberlite groundmass (1.2–6.1 ppm and 5.2–46 ppm for bulk samples in this study; Supplementary Data 3) provides ample opportunity for implantation of $^4He*$ into olivine (as previously demonstrated for olivines in Jurassic basalts[88]). The amount ($I$) of radiogenic $^4He*$ implanted into olivine fluid inclusions from the kimberlite groundmass was calculated using the equation of Dunai and Wijbrans[89].

$$I = {}^4He^* \times \left( \frac{3 \times S}{4 \times R} + \frac{S^3}{16 \times R^3} \right) \quad (2)$$

Where $S$ represents the track length (~20 μm) of alpha particles in kimberlite and olivine and $R$ is the olivine radius (1000 μm). To calculate $^4He*$ produced in the kimberlite groundmass, we used Eq. (1), kimberlite emplacement ages (Supplementary Data 1) and groundmass U-Th contents equal to twice the U and Th contents of the bulk kimberlite (Supplementary Data 3) to account for the diluting effect of ~50% by U-Th-free olivine in kimberlites. Similar results were obtained using the formulation of Lal[90].

$$I = {}^4He^* \times \frac{3 \times S}{4 \times R} \times \frac{\rho_k}{\rho_{ol}} \quad (3)$$

where $\rho_k$ and $\rho_{ol}$ are the densities of kimberlite (2.5 g/cm$^3$) and olivine (3.3 g/cm$^3$), respectively. Implanted $^4He*$ can be similar or exceed measured $^4He$ even assuming an arbitrary 10% of retention of $^4He*$ to account for the fraction of implanted $^4He*$ that gets trapped in fluid

inclusions (Supplementary Fig. S6). The effect of $^4$He* implantation is probably most substantial in the Cambrian and Neoproterozoic kimberlites, which all show low $^3$He/$^4$He coupled with elevated $^4$He contents (Fig. 2), and perhaps also in low-$^3$He/$^4$He Mesozoic kimberlites from Koidu and Tonguma, which exhibit the highest concentrations of U (4.7–6.1 ppm) and Th (43–46 ppm) due to their mica-rich groundmass. However, these calculations provide upper estimates because olivine rims are not always fully preserved in kimberlites due to serpentinisation (Fig. 1). In addition, the fraction of implanted $^4$He* that is trapped in the fluid inclusions is unknown, which makes this model inconclusive.

In summary, while both radiogenic ingrowth and implantation act to lower the $^3$He/$^4$He of olivine compared to the original values at the time of kimberlite crystallisation, the effect of these processes cannot be accurately quantified. Radioactive decay and implantation are of marginal relevance for Ne isotopes because track lengths of $^{21}$Ne* are much shorter than for $^4$He*, nucleogenic production of Ne is much slower and Ne diffusion slower than for He[17].

## He-Ne isotope mixing models

Binary mixing calculations were undertaken for $^4$He/$^3$He and $^{21}$Ne/$^{22}$Ne to simulate contamination of exsolved kimberlite fluids by continental crust, interaction between kimberlite melt and metasomatised lithospheric mantle, and contribution of recycled oceanic crust to the kimberlite source (Fig. 7). The compositions employed for each of the mixing endmembers are summarised in Supplementary Data 4 and described below.

For the kimberlite source, two different sets of isotopic compositions are employed for modelling and correspond to sources in the upper mantle (similar to MORB[1]: $^4$He/$^3$He = $9.0 \times 10^4$; $^{21}$Ne/$^{22}$Ne = 0.060) and low-$^3$He/$^4$He lower mantle (Baffin or proto-Icelandic plume[6,16]: $^4$He/$^3$He = $1.1 \times 10^4$; $^{21}$Ne/$^{22}$Ne = 0.035), respectively. The contents of He and Ne in the lower mantle are considered to be similar to those of mantle xenoliths enriched in noble gases ($1 \times 10^{-5}$ cm$^3$/g and $1 \times 10^{-10}$ cm$^3$/g, respectively) with half of these concentrations but the same He/Ne ratio employed for the more degassed upper mantle. Using a higher Ne/He ratio in the plume source compared to the upper mantle[19,20] would not substantially alter the results of these models. These same He-Ne isotopic compositions are employed for kimberlite melt and exsolved fluid, but the concentrations of He and Ne are assumed to be about two orders of magnitude higher than in the source to account for the strongly incompatible nature of noble gases. For the kimberlite melt endmember, He and Ne contents are similar to those of the Atlantic "popping rocks" with the highest noble gas concentrations[52,63]: He = $1 \times 10^{-4}$ cm$^3$/g; Ne = $3 \times 10^{-9}$ cm$^3$/g. The rationale for this choice is that noble gases in volatile-poor tholeiites (MORBs) were shown to have similar solubilities as in carbonate-rich melts (kimberlites)[57] perhaps because higher $CO_2$ contents decrease noble gas solubilities in (silicate) melts[91]. For the exsolved kimberlite fluid, the He content is halved compared to that in the kimberlite melt to account for the lower solubility of He compared to Ne in C-O-H fluids[20,57,58].

The He content of the continental crust contaminant is assumed to be $1 \times 10^{-3}$ cm$^3$/g based on typical U-Th contents of the upper crust[74] and an age of 2 Ga, with the Ne concentration ($1 \times 10^{-8}$ cm$^3$/g) obtained using the employed He content and typical He/Ne ratios for crustal fluids[92]. $^4$He/$^3$He ($7.5 \times 10^7$) and $^{21}$Ne/$^{22}$Ne (0.52) for the continental crust are taken from the compilation of Ballentine and Burnard[60].

The metasomatised lithospheric mantle is considered to have the same Ne contents as the mantle xenoliths with the highest noble gas concentrations (Ne = $1 \times 10^{-10}$ cm$^3$/g, that is $^{22}$Ne ~ $1 \times 10^{-11}$ cm$^3$/g) and higher He contents than mantle xenoliths (i.e. He = $1 \times 10^{-5}$ cm$^3$/g) resulting from radiogenic ingrowth of $^4$He* in phlogopite-bearing lithologies with elevated U and Th contents (e.g., $4.6 \times 10^{-5}$ cm$^3$/g using

U-Th contents from McIntyre et al.[93], and an age of 2 Ga). The He-Ne isotopic composition is varied between present-day estimates ($^4$He/$^3$He = $1.2 \times 10^5$; $^{21}$Ne/$^{22}$Ne = 0.071) from Gautheron et al.[94] and higher $^4$He/$^3$He and $^{21}$Ne/$^{22}$Ne ratios that account for high time-integrated U and Th contents resulting from ancient metasomatic enrichment (e.g., $^4$He/$^3$He = $5.6 \times 10^5$; $^{21}$Ne/$^{22}$Ne = 0.237 for 2 Ga; Supplementary Data 4).

Finally, we model the age-dependent He-Ne isotopic composition of subducted oceanic crust using the following parameters: initial He = $1 \times 10^{-7}$ cm$^3$/g, and Ne = $1 \times 10^{-11}$ cm$^3$/g (or $^{22}$Ne ~ $1 \times 10^{-12}$ cm$^3$/g) corresponding to 1% and 10%, respectively, of He and Ne measured in typical MORBs and oceanic gabbros to account for He and Ne loss during subduction dehydration; $^4$He/$^3$He = $1.2 \times 10^5$ and $^{21}$Ne/$^{22}$Ne = 0.029 (that is air-like Ne) as measured by Moreira et al.[66] for partially altered oceanic gabbros. The age of subducted crust is varied between 1 and 2 Gyr and ingrowth of radiogenic $^4$He* and nucleogenic $^{21}$Ne* are calculated using U and Th concentrations of subducted-modified oceanic crust (0.029 and 0.252 ppm, respectively)[95]. Only 10% of $^4$He* is retained in the subducted oceanic crust to account for diffusive loss of $^4$He to the ambient mantle[96]. However, if the resulting He concentration of the subducted crustal component becomes lower than He in the kimberlite mantle source, the latter is employed for the subducted crust (because diffusion is prevented due to lack of a He concentration gradient).

While we strive to employ endmembers with compositions as close as possible to natural values, we note that, especially for He and Ne concentrations, no independent constraints exist. Therefore, the binary mixing models should be considered approximations of natural scenarios.

## Additional discussion of He-Ne isotope data

Some olivine separates provide anomalous Ne isotope compositions or the data appear to be inconclusive and are hence discussed here rather than in the main text. Jericho (Canada) and Karowe (Botswana) olivines represent special cases in this study where linear regressions through the Ne isotope data do not intersect the air value but rather a "fractionated air" component that probably consists of a mixture of air and crustal Ne (Fig. 3). Although the slopes of regressions through these data are similar to or shallower than that of MORBs, it is not possible to discern potential lower mantle contributions (if present) in these cases due to inputs from multiple components and lack of meaningful $^{21}$Ne/$^{22}$Ne$_S$.

Of the two analyses of olivines from the Igwisi Hills (Tanzania), one shows atmospheric Ne and the second one a composition similar to those of the Lac de Gras olivines (Figs. 3 and 7). However, with only one available point distinct from air-Ne, any interpretation of these data is necessarily weak. Finally, for Letseng only one Ne isotope analysis is available (i.e. $^{20}$Ne > $6 \times 10^{-12}$ cc): in this case, $^{21}$Ne/$^{22}$Ne$_S$ cannot be considered robust because the second component could be "fractionated air" as per the olivines from Jericho and Karowe. These considerations help explain why the Karowe and Letseng analyses plot at anomalously high $^{21}$Ne/$^{22}$Ne$_S$ considered their relatively low $^4$He/$^3$He in Fig. 7 where no mixing model can be produced.

Direct correlations observed between log($^4$He/$^{40}$Ar*) and log($^4$He/$^{21}$Ne*) in kimberlite olivine (Fig. 6) suggest variable extents of He loss or trapping of exsolved fluids. Helium loss and/or fluid exsolution must have also impacted $^3$He/$^{22}$Ne$_S$ (where $^{22}$Ne$_S$ reflect extrapolation to a $^{20}$Ne/$^{22}$Ne of 12.5; Supplementary Fig. S7). Kimberlite olivines plot along a sub-parallel trend in log($^3$He/$^{22}$Ne$_S$) vs log($^4$He/$^{21}$Ne*) which extends to low $^4$He/$^{21}$Ne* at relatively invariant $^3$He/$^{22}$Ne$_S$, at odds with data from MORBs and OIBs. The origin of these low $^3$He/$^{22}$Ne$_S$ in kimberlites (1.0–4.2) is unclear and might reflect contributions from fluid exsolution as well as crustal contamination, which all contribute to lower $^3$He/$^{22}$Ne from values perhaps as high as those of MORBs or Baffin picrites, i.e. the proto-Icelandic plume ( ~10)[6,97].

## Data availability

All the data generated in this study are provided in the Supplementary Information.

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

## Acknowledgements

Samples from Maniitsoq in Greenland were collected as part of a mapping project of the Ministry of Mineral Resources, Government of Greenland, and we thank all members of the project who contributed to the fieldwork. Max Schmidt, Richard Brown, Hugh O'Brien and Yaakov Weiss are thanked for providing the material analysed from Letseng, Igwisi Hills, Finland Pipe 1 and Ghacho Kué, respectively. De Beers Group kindly provided the examined sample from the Victor mine. This project was funded by the Swiss National Foundation (Ambizione fellowship no. PZ00P2_180126/1 to A.G.).

## Author contributions

A.G. designed the study and wrote the original manuscript draft. M.K., P.H.B., J.M.C. and F.M.S. contributed to the noble gas analyses. A.G. and S.O. analysed the data. Q.C., B.J.P. and J.M.K. performed the trace element and radiogenic isotope analyses. A.G., K.S. and D.G.P. provided the samples. MK, P.H.B., F.M.S., S.O., B.J.P., K.S. and D.G.P. contributed to manuscript editing and revision.

## Competing interests

The authors declare no competing interests.
