## [Peer Review File · Nature Communications]

Primordial neon and the deep mantle origin of kimberlites

Corresponding Author: Dr Andrea Giuliani

Version 0:

Reviewer comments:

Reviewer #1

(Remarks to the Author)

The authors present novel noble gas data (He-Ne-Ar) alongside additional geochemical information, though the latter is not extensively explored in this work, as the focus is primarily on noble gases. The study examines kimberlite samples from various locations, with a key result being the evidence for primordial neon, similar to that observed in certain hotspots such as Kerguelen. In contrast, helium displays a subcontinental mantle signature, highlighting a divergence between He and Ne systematics, a phenomenon that has been documented in prior studies.

The paper is well-written, although it caters to a specialist audience, particularly those with expertise in noble gases. The main message, such as the detection of primordial neon, is somewhat obscured by detailed calculations that are commonly applied to interpret noble gas features in older samples. Nevertheless, I find the paper remarkable, especially considering the analytical challenges associated with He-Ne-Ar measurements in olivines extracted from kimberlites.

While I have no major criticisms of the manuscript, I do have one suggestion that may challenge the current interpretation regarding the primitive nature of the mantle source in some kimberlites. The formation of these samples might involve kinetic fractionation of neon isotopes. In the accompanying document, I propose a theoretical isotopic fractionation model based on Graham's law, starting from an intermediate "MORB-like" composition, assuming that MORB-air mixing occurs before this fractionation. If such a process took place, it could explain the presence of non-nucleogenic neon in some samples, without invoking a primitive mantle source. This would align with the helium systematics, which show no evidence for primitive helium.

Although this suggestion remains theoretical, I believe it warrants discussion in the manuscript. Furthermore, the inclusion of another isotopic system, such as the $^{38}\text{Ar}/^{36}\text{Ar}$ ratio, could provide additional insights; unfortunately, this ratio was not measured in the study.

In conclusion, I recommend the manuscript for publication, but with minor to major revisions to incorporate a discussion of the potential role of kinetic isotopic fractionation in the neon data interpretation.

Reviewer #2

(Remarks to the Author)

The authors have produced a high quality He-Ne-Ar isotope dataset from kimberlite olivine separates. Combined with high quality Sr-Nd-Hf isotope measurements and trace element compositions of the same olivine separates, the authors have used the data to investigate the likely source(s) of kimberlites. The samples were selected to represent a wide range of kimberlitic material in terms of both geographic and temporal distribution.

The results, in the main, confirm the findings of previous similar studies and add to the number of measurements made of noble gas isotopes in kimberlites, the neon isotope measurements being a particular welcome addition. As to new contributions to the literature, while most of the Ne isotope measurements are novel, I'm not convinced that the contribution as a whole is the panacea to the problems associated with kimberlite petrogenesis that the authors allude to in the title and abstract. In fact, of the samples studied, very few seem to preserve Ne isotope evidence that unequivocally roots the origin of kimberlites firmly in the deep mantle. The suggestion that Ne might supplant He as the "go to" tracer for kimberlite genesis is tantalising but, again, the limited number of samples that yield incontrovertible evidence for this means that I remain not entirely convinced that Ne can necessarily solve this enigma.

I have no problems at all with the methods employed, or the quality of the data produced. In this respect, if someone else wanted to pursue this line of enquiry further then they have a transparent route map of how to embark on such a journey. While the authors suggest that further investigation of more samples might be a fruitful endeavour, capable of shedding light on the problems that they attempted to solve, it strikes me that uncertainties involved, despite involved and detailed modelling, this might not be the toolbox to deploy for this job.

Below I have added so specific comments according to the line numbers associated with the Word document downloaded from the Editorial site. I hope the authors find them useful.

Abstract

Lines 38 to 40: It's a bold statement to effectively oust He isotopes as a key tracer of deep-mantle contributions. Having read the manuscript, I agree that it has some use in a limited number of examples, but I don't think it can supplant He just yet.

Introduction

Line 48: Is it worth adding the list of noble gases in brackets here?

Line 49 to 54: This is a long sentence that contains multiple, bracketed subsidiary statements. It would benefit from being broken down into more manageable chunks. For what it's worth, my own rule of thumb is that if I can't read it out loud in a single breath (at sensible speed!), the sentence is too long.

Lines 54 to 57: Again, it's another long sentence that needs breaking down into at least two shorter ones.

Lines 57 to 61: Another whopper of a sentence. By the time I'm getting to the end of these mammoth sentences I'm forgetting how they started.

Line 64: Does Nature Communications insist on defining abbreviations at first usage? The same applies for MORB in the next paragraph. LIP and LLSVP are defined, so I guess this needs to be made consistent.

Lines 66 to 67: Is there a key reference that could be added here to support this statement?

Lines 69 to 70: Why not simply show the range of MORB values rather than adding on the qualifier at the end, which ultimately drags out the sentence?

Lines 71 to 74: This could quite easily be split into two sentences to make it more easily read.

Lines 90 to 95: This needs breaking down into two shorter sentences.

Lines 113 to 116: It might be worth pointing out here that refs 22 and 39 were both published in the same year - the style of referencing employed in this journal does not make that obvious, but this would emphasise the point being made.

Line 123: Why mention just these two? You could just direct the reader to the supplementary table for the full list, as you do immediately after.

Lines 127 to 132: Another long sentence here.

Results

Lines 149 to 150: I find it odd to include absent phases in a sentence describing the contents of the kimberlites - this could be worded differently.

Line 185: "...generally resolvable..." doesn't fill me with confidence. Are they resolvable or not?...and if only some are what proportion are resolvable?

Lines 190 to 192: How this is worded makes this look incremental, i.e. it slightly extends the range of values for this locality.

Lines 193 to 194: Is this a different finding to that of Sumino et al? If not, then I'm struggling to see the contribution here.

Line 214: It isn't clear to me to what fractions relate.

Line 217: Here and throughout - I don't know what Nature's policy is on units of concentration, but ppm is lab slang for micrograms per gram.

Lines 221 to 223: What is the effect of this surface contamination on the noble gas abundances and their isotope ratios?

Discussion

Lines 241 to 242: Aren't these two samples the ones whose olivines show that the trapped fluids preserve evidence of crustal contamination? (two sentences earlier). Doesn't that add uncertainty to the interpretation of the noble gas data that follows?

Line 241 to 261: I find this whole opening section to the discussion really quite speculative.

Lines 273 to 275: This gives the impression that the nature of the inclusions in the olivine isn't really that clear - the authors are not able to distinguish between fluid inclusions and melt inclusions. Isn't that problematic when it comes to interpreting the significance of the composition of the inclusions?

Line 307: I didn't see any Hf data in these figures. If it doesn't demonstrate anything that doesn't make it into the main text, does it need to be included at all?

Line 376: Aren't the concentrations of LREE high in the crust? What sort of crust are we discussing here?

Implications

Lines 387 to 399: This first paragraph of Implications seems very clear compared to what has preceded it. However, this seems mainly to be a particularly lucid review of previous studies that would be helpful for a general reader much earlier in the manuscript.

Line 401: But this is only 2 out of the 18 studied. Does this really point to a ubiquitous process?

Lines 413 to 418: In reality, could the authors see anyone taking on a study like the hypothetical one they are proposing here, especially in the light of so many of the 18 samples studies here generating results that do not shed much light on the "source of kimberlites" problem? I don't think that this sentence really achieves anything and can be removed with no detriment to the rest of the manuscript.

Lines 420 to 425: I think that the novelty of this study is the utility of using Ne isotopes to examine the likelihood of deep-mantle sources for kimberlites, but even this has only identified a few kimberlites where the technique has revealed anything that could be described as unequivocal. The fact that the authors choose to relegate discussion of the remaining samples to the supplementary material suggests that while the data is of a high quality, it's interpretation is problematic and the use of Ne isotopes in this context might not be the panacea that they'd like it to be.

Figures

Figure 3: Why show only 1 sigma uncertainties?

Figure 4 (and main text commentary that refers to it): So 10/21 (nearly half) of the trace elements here are associated with some anomaly or other attributed to a non-magmatic origin. The main text also states that some olivine separates were visibly contaminated with bulk-rock matrix. Beyond the general observation that those elements that are more incompatible during partial melting are more abundant than those that are less incompatible, I'm not as convinced of this irrefutable magmatic origin as the authors seem to be.

Version 1:

Reviewer comments:

Reviewer #1

(Remarks to the Author)

I have carefully reviewed the revised version of the manuscript and find that the authors have addressed the comments in a thorough and convincing manner. I have no further comments or suggestions, and I recommend accepting the manuscript in its current form.

Reviewer #2

(Remarks to the Author)

Having had the opportunity to look at the revised manuscript, I would like to thank the authors for engaging with my comments and queries so positively and comprehensively. There were clearly a couple of instances where I had

misunderstood a key point and these have been clarified for me either by rewording some of the text or explaining more clearly where I had misunderstood the significance of some of the findings.

As it stands, I have no problem with recommending that the article in its current form be published in Nature Communications. It is a valuable contribution to the literature that will have a broad appeal among the geoscience community

REVISION NOTES

Reviewers' comments:

Reviewer #1 (Remarks to the Author):

The authors present novel noble gas data (He-Ne-Ar) alongside additional geochemical information, though the latter is not extensively explored in this work, as the focus is primarily on noble gases. The study examines kimberlite samples from various locations, with a key result being the evidence for primordial neon, similar to that observed in certain hotspots such as Kerguelen. In contrast, helium displays a subcontinental mantle signature, highlighting a divergence between He and Ne systematics, a phenomenon that has been documented in prior studies.

The paper is well-written, although it caters to a specialist audience, particularly those with expertise in noble gases. The main message, such as the detection of primordial neon, is somewhat obscured by detailed calculations that are commonly applied to interpret noble gas features in older samples. Nevertheless, I find the paper remarkable, especially considering the analytical challenges associated with He-Ne-Ar measurements in olivines extracted from kimberlites.

We are glad that Dr Moreira found our paper "remarkable". We would like to clarify that we did not make extensive use of the wealth of additional geochemical data generated for this study because our primary goal was to employ those data to understand whether or not the fluid inclusions in olivine, which host the noble gases, represent pristine magmatic fluids. We do not entirely agree that the paper is exclusively directed to noble-gas specialists. It has profound implications for the genesis of kimberlites, a topic of interest to the wider deep Earth-science community as evidenced by the relatively high citation rate of kimberlite studies.

While I have no major criticisms of the manuscript, I do have one suggestion that may challenge the current interpretation regarding the primitive nature of the mantle source in some kimberlites. The formation of these samples might involve kinetic fractionation of neon isotopes. In the accompanying document, I propose a theoretical isotopic fractionation model based on Graham's law, starting from an intermediate "MORB-like" composition, assuming that MORB-air mixing occurs before this fractionation. If such a process took place, it could explain the presence of non-nucleogenic neon in some samples, without invoking a primitive mantle source. This would align with the helium systematics, which show no evidence for primitive helium. Although this suggestion remains theoretical, I believe it warrants discussion in the manuscript.

This is an excellent suggestion and we now explicitly acknowledge this alternative interpretation in the revised manuscript (lines 379-384) and added a vector showing the effect of kinetic fractionation from typical MORB compositions in Figure 3. However, we do not favour this interpretation for the following reasons: 1) only the two samples with the most geochemically-depleted isotopic signature (i.e. highest Nd isotopes) exhibit Ne isotopes that are less nucleogenic than typical MORB compositions. These samples are those which have been least affected by subducted material in the source and hence are most likely to preserve deep and/or ancient noble-gas signatures as those typical of

some mantle plumes; 2) for these kimberlites there is additional evidence of plume contribution based on location along the Great Meteor hot-spot track (Victor), and He and ^{182}W isotopes (Udachnaya) (lines 369-376); and 3) if kinetic fractionation occurred after MORB-air mixing, it would be a remarkable coincidence that all the new analyses of the Udachnaya samples, which are olivine separates from different specimens, are aligned along a regression which intersects air (see Figure 3). In addition, olivine from this kimberlite previously analysed by Sumino et al. (2006) also display similarly primordial Ne isotope composition.

Furthermore, the inclusion of another isotopic system, such as the $^{38}\text{Ar}/^{36}\text{Ar}$ ratio, could provide additional insights; unfortunately, this ratio was not measured in the study. The review is correct. ^{38}Ar was not measured and hence Ar isotopes are not discussed in detail in this manuscript.

In conclusion, I recommend the manuscript for publication, but with minor to major revisions to incorporate a discussion of the potential role of kinetic isotopic fractionation in the neon data interpretation.

Reviewer #2 (Remarks to the Author):

The authors have produced a high quality He-Ne-Ar isotope dataset from kimberlite olivine separates. Combined with high quality Sr-Nd-Hf isotope measurements and trace element compositions of the same olivine separates, the authors have used the data to investigate the likely source(s) of kimberlites. The samples were selected to represent a wide range of kimberlitic material in terms of both geographic and temporal distribution.

The results, in the main, confirm the findings of previous similar studies and add to the number of measurements made of noble gas isotopes in kimberlites, the neon isotope measurements being a particular welcome addition.

[and]

Lines 190 to 192: How this is worded makes this look incremental, i.e. it slightly extends the range of values for this locality.

[and]

Lines 193 to 194: Is this a different finding to that of Sumino et al? If not, then I'm struggling to see the contribution here.

The main purpose of this study was to undertake a thorough analysis of kimberlites with variable geochemical and isotopic features where unaltered olivine was available – this is no mean feat. We certainly have re-examined Udachnaya-East and obtained similar results as Sumino et al. (2006; Figures 2 and 3), which is an indication that our analytical protocols can identify deep mantle contributions in Paleozoic kimberlites. Conversely, our new dataset disproves the previous suggestion of strongly unradiogenic He isotopes in West Greenland kimberlites from Sarfartoq (Figure 2). In addition, our

global dataset which combines isotopes of noble gases with lithophile elements, shows which kimberlites contain plume-like isotopic signatures (Figure 7). These results pave the way towards the examination of geochemically-depleted kimberlites (i.e. those least enriched by deeply subducted and/or metasomatised lithospheric material) to investigate the preservation of ancient mantle heterogeneities in the deep Earth (lines 42-45 and 435-441).

As to new contributions to the literature, while most of the Ne isotope measurements are novel, I'm not convinced that the contribution as a whole is the panacea to the problems associated with kimberlite petrogenesis that the authors allude to in the title and abstract. In fact, of the samples studied, very few seem to preserve Ne isotope evidence that unequivocally roots the origin of kimberlites firmly in the deep mantle. The suggestion that Ne might supplant He as the "go to" tracer for kimberlite genesis is tantalising but, again, the limited number of samples that yield incontrovertible evidence for this means that I remain not entirely convinced that Ne can necessarily solve this enigma.

As noted by the reviewer in the first paragraph of his/her review, we have examined a large number of samples (35 olivine separates from 18 kimberlites) with broad temporal and geographic distribution, without expectations of which samples could provide plume-like signatures. It is remarkable that the only two kimberlites with a clear plume signature are those with the most geochemically-depleted isotopic compositions (i.e. highest Nd isotopes). The implication of this observation is that pristine noble-gas signatures of the deep and/or ancient mantle are limited to the kimberlites whose source is least affected by addition of recycled subducted material or metasomatic lithospheric components. We believe this is an extraordinary finding, which contrasts with other mantle-derived magmas such as oceanic island basalts where primordial noble-gas signatures are also observed in geochemically enriched samples. These insights are now better stressed in the manuscript (lines 362-369 and 421-423).

I have no problems at all with the methods employed, or the quality of the data produced. In this respect, if someone else wanted to pursue this line of enquiry further then they have a transparent route map of how to embark on such a journey. While the authors suggest that further investigation of more samples might be a fruitful endeavour, capable of shedding light on the problems that they attempted to solve, it strikes me that uncertainties involved, despite involved and detailed modelling, this might not be the toolbox to deploy for this job.

We have clarified why we believe this pioneering study of noble-gas geochemistry in kimberlites sets the stage for future work, including the application of noble-gas and other isotopic tracers of deep and/or ancient mantle (e.g., ^{129}Xe , ^{182}W , ^{142}Nd) to better evaluate the occurrence of early-Earth heterogeneities in the source of kimberlites (lines 42-45 and 435-441).

Below I have added so specific comments according to the line numbers associated with the Word document downloaded from the Editorial site. I hope the authors find them useful.

We would like to thank this reviewer for the thorough and detailed evaluation of our work. His/her comments have helped improve the manuscript contents and clarity. We hope that our additions clarify why we believe this work will have a major impact in understanding the origin of kimberlites, including which samples should be targeted in future studies.

Abstract

Lines 38 to 40: It's a bold statement to effectively oust He isotopes as a key tracer of deep-mantle contributions. Having read the manuscript, I agree that it has some use in a limited number of examples, but I don't think it can supplant He just yet.

We are tried to tone down this statement by making it specific to "intraplate continental magmas" (line 44) rather than intraplate magmas in general. However, we note that the statement indicates that Ne isotopes appear to be "more robust tracers of deep-mantle contributions" than He isotopes, hence it does not oust He, but just reinforces what some other studies have also noted.

Introduction

Line 48: Is it worth adding the list of noble gases in brackets here?

Done (line 54).

Line 49 to 54: This is a long sentence that contains multiple, bracketed subsidiary statements. It would benefit from being broken down into more manageable chunks. For what it's worth, my own rule of thumb is that if I can't read it out loud in a single breath (at sensible speed!), the sentence is too long.

Done (lines 56-60).

Lines 54 to 57: Again, it's another long sentence that needs breaking down into at least two shorter ones.

We prefer to retain this sentence, as we did not find it excessively long.

Lines 57 to 61: Another whopper of a sentence. By the time I'm getting to the end of these mammoth sentences I'm forgetting how they started.

Sentence split into two sentences (lines 63-68).

Line 64: Does Nature Communications insist on defining abbreviations at first usage? The same applies for MORB in the next paragraph. LIP and LLSVP are defined, so I guess this needs to be made consistent.

OIB and MORB were defined at lines 56 and 59, respectively.

Lines 66 to 67: Is there a key reference that could be added here to support this statement?

We have added references to Kurz et al. (1982), Graham (2002) and Mukhopadhyay (2012) – references 1-3.

Lines 69 to 70: Why not simply show the range of MORB values rather than adding on the qualifier at the end, which ultimately drags out the sentence?

We have removed the qualifier to simplify this statement (line 76).

Lines 71 to 74: This could quite easily be split into two sentences to make it more easily read.

Done (lines 77-80).

Lines 90 to 95: This needs breaking down into two shorter sentences.

Done (lines 97-102).

Lines 113 to 116: It might be worth pointing out here that refs 22 and 39 were both published in the same year - the style of referencing employed in this journal does not make that obvious, but this would emphasise the point being made.

We are not sure how we can point this out considering the referencing style of the journal. We are also not sure why this information is considered so critical by the reviewer.

Line 123: Why mention just these two? You could just direct the reader to the supplementary table for the full list, as you do immediately after.

Because these kimberlites were previously shown to contain He and Ne isotopic signatures typical of deep-mantle plumes as indicated in the previous paragraph (lines 118-123).

Lines 127 to 132: Another long sentence here.

Done (lines 134-139).

Results

Lines 149 to 150: I find it odd to include absent phases in a sentence describing the contents of the kimberlites - this could be worded differently.

Reworded (lines 156-157).

Line 185: "...generally resolvable..." doesn't fill me with confidence. Are they resolvable or not?...and if only some are what proportion are resolvable?

Rephrased to "All Ne isotope ratios except for two measurements are resolvable ($>2 \times$ standard deviation) from the air composition" (lines 192-194).

Line 214: It isn't clear to me to what fractions relate.

Rephrased as "This inference is confirmed by the comparison of laser ablation and solution-mode ICP-MS analyses of mantle olivine containing secondary fluid inclusions, which show eight- and seven-time higher concentrations in solution-mode analyses for Sr and U, respectively." (lines 220-223).

Line 217: Here and throughout - I don't know what Nature's policy is on units of concentration, but ppm is lab slang for micrograms per gram.

Happy to change ppm to $\mu\text{g/g}$ if required by the editor.

Lines 221 to 223: What is the effect of this surface contamination on the noble gas abundances and their isotope ratios?

This is addressed in the Methods section “Olivine selection and assessment of sample contamination” and shown graphically in Supplementary Figure S3 for He isotopes.

Discussion

Lines 241 to 242: Aren't these two samples the ones whose olivines show that the trapped fluids preserve evidence of crustal contamination? (two sentences earlier).

Doesn't that add uncertainty to the interpretation of the noble gas data that follows?

The review is correct. The fluids hosted in olivine from Victor and Udachnaya-East contain a crustal component based on their Sr isotopes (Figure 5). The implication for noble-gas isotopes is that the measured compositions represent minimum values for $^3\text{He}/^4\text{He}$, whereas the influence on Ne isotopes is probably negligible based on our modelling (lines 386-393 and Figure 7).

Line 241 to 261: I find this whole opening section to the discussion really quite speculative.

The introductory paragraph of the Discussion essentially sets the scene for the rest of the discussion while addressing some of the complicating factors associated with the interpretation of noble-gas isotopes in old rocks. We are not sure what is speculative here – perhaps the model outlining $^4\text{He}^*$ ingrowth and implantation, which is detailed in the Methods section (lines 257-262)? We would be happy to improve the paragraph if the reviewer could be more specific.

Lines 273 to 275: This gives the impression that the nature of the inclusions in the olivine isn't really that clear - the authors are not able to distinguish between fluid inclusions and melt inclusions. Isn't that problematic when it comes to interpreting the significance of the composition of the inclusions?

The reviewer is correct about the lack of clarity on the origin of fluid inclusions in kimberlite olivine, with some Authors considering them carbonate-rich melts rather than C-O-H fluid. The issue here is that there is a physical continuum between carbonatitic melts rich in H_2O and volatiles, and high-T aqueous-carbonic fluids rich in dissolved alkalis and metals (e.g., Yuan et al., 2023 Science Advances). This issue is negligible for the He-Ne-Ar isotopic signature of crushed olivine, but has implications for the correct interpretation of noble-gas abundance ratios (He/Ne, He/Ar). However, a correct interpretation of noble-gas abundance ratios is not critical to this manuscript (lines 281-284).

Line 307: I didn't see any Hf data in these figures. If it doesn't demonstrate anything that doesn't make it into the main text, does it need to be included at all?

The Hf isotope data are in Supplementary Figure S5, a reference to which is now included in the text (line 316).

Line 376: Aren't the concentrations of LREE high in the crust? What sort of crust are we discussing here?

We have amended the text to clarify that this is average continental crust (line 391), which has low LREE concentrations compared to kimberlites.

Implications

Lines 387 to 399: This first paragraph of Implications seems very clear compared to what has preceded it. However, this seems mainly to be a particularly lucid review of previous studies that would be helpful for a general reader much earlier in the manuscript.

This section combines our new data with those available for mantle xenoliths to conclude that 1) the source of noble gases in mantle xenoliths is dominantly represented by the entrained magmas; and 2) the noble-gas signature of kimberlites and other intraplate magmas is not substantially influenced by interaction with lithospheric-mantle wall rocks. We disagree this is a simple review and hence would like to retain this paragraph as it is.

Line 401: But this is only 2 out of the 18 studied. Does this really point to a ubiquitous process?

It does not, which is indeed not intended to be the message here. Hence the wording "some kimberlites, and specifically those least affected by addition of deeply subducted material and/or interaction with metasomatised lithospheric-mantle rocks (e.g., Victor, Udachnaya-East), contain primordial Ne derived from the lower mantle" (lines 421-423). The paragraph also clearly states that other kimberlites "are not related to deep-mantle plumes as shown by the noble-gas data presented herein" (lines 430-435).

Lines 413 to 418: In reality, could the authors see anyone taking on a study like the hypothetical one they are proposing here, especially in the light of so many of the 18 samples studies here generating results that do not shed much light on the "source of kimberlites" problem? I don't think that this sentence really achieves anything and can be removed with no detriment to the rest of the manuscript.

We understand the reviewer concern and have modified the manuscript to acknowledge that additional analyses of noble-gas isotopes as well as other tracers of early-Earth processes including ^{129}Xe , ^{182}W , ^{142}Nd should address geochemically-depleted kimberlites which have been least affected by addition of deeply subducted material, interaction with metasomatised lithospheric-mantle rocks and/or crustal contamination (lines 435-441).

Lines 420 to 425: I think that the novelty of this study is the utility of using Ne isotopes to examine the likelihood of deep-mantle sources for kimberlites, but even this has only identified a few kimberlites where the technique has revealed anything that could be described as unequivocal. The fact that the authors choose to relegate discussion of the remaining samples to the supplementary material suggests that while the data is of a high quality, its interpretation is problematic and the use of Ne isotopes in this context might not be the panacea that they'd like it to be.

We are glad the reviewer appreciates the novelty of this approach. On the other hand, they didn't note that approximately half of the discussion (lines 301-353) is dedicated to samples which do not show plume-like Ne isotopes. The additional discussion of the remaining noble-gas data, which the reviewer refers to, is just 356 words. This discussion is relegated to the Methods section because it is of secondary importance and, in part, more controversial while not being central to the manuscript.

Figures

Figure 3: Why show only 1 sigma uncertainties?

This is common practice for Ne isotopes.

Figure 4 (and main text commentary that refers to it): So 10/21 (nearly half) of the trace elements here are associated with some anomaly or other attributed to a non-magmatic origin. The main text also states that some olivine separates were visibly contaminated with bulk-rock matrix. Beyond the general observation that those elements that are more incompatible during partial melting are more abundant than those that are less incompatible, I'm not as convinced of this irrefutable magmatic origin as the authors seem to be.

If the fluids were not dominantly magmatic, the trace element patterns would substantially differ from those of the kimberlite host and resemble the continental crust, at least locally. This is not the case (Figure 4). We hope the reviewer agrees. We have not included additional details to explain, for example, the positive anomalies for Ti and Ta observed in some olivines, which probably reflect (optically visible) inclusions of Fe-Ti oxide minerals. These details do not affect the main conclusions of this manuscript but would rather dilute the key observations and related interpretations.